# Finding Actual Descent Directions for Adversarial Training

**Fabian Latorre**[*], **Igor Krawczuk**[*], **Leello Dadi**[*], **Thomas Pethick** and **Volkan Cevher**
EPFL, Switzerland
`firstname.lastname@epfl.ch`

## Abstract

Adversarial Training using a strong first-order adversary (PGD) is the gold standard for training Deep Neural Networks that are robust to adversarial examples. We show that, contrary to the general understanding of the method, the gradient at an optimal adversarial example may increase, rather than decrease, the adversarially robust loss. This holds independently of the learning rate. More precisely, we provide a counterexample to a corollary of Danskin's Theorem presented in the seminal paper of Madry et al. (2018) which states that a solution of the inner maximization problem can yield a descent direction for the adversarially robust loss. Based on a correct interpretation of Danskin's Theorem, we propose Danskin's Descent Direction (DDi) and we verify experimentally that it provides better directions than those obtained by a PGD adversary. Using the CIFAR10 dataset we further provide a real world example showing that our method achieves a steeper increase in robustness levels in the early training stages of smooth-activation networks without BatchNorm, and is more stable than the PGD baseline. As a limitation, PGD training of ReLU+BatchNorm networks still performs better, but current theory is unable to explain this.

## 1    Introduction

Adversarial Training (AT) (Goodfellow et al., 2015; Madry et al., 2018) has become the de-facto algorithm used to train Neural Networks that are robust to adversarial examples (Szegedy et al., 2014). Variations of AT together with data augmentation yield the best-performing models in public benchmarks (Croce et al., 2020). Despite lacking optimality guarantees for the inner-maximization problem, the simplicity and performance of AT are enough reasons to embrace its heuristic nature.

From an optimization perspective, the consensus is that AT is a sound algorithm: based on Danskin's Theorem, Madry et al. (2018, Corollary C.2) posit that by finding a maximizer of the inner non-concave maximization problem, i.e., an optimal adversarial example, one can obtain a descent direction for the adversarially robust loss. *What if this is not true? are we potentially overlooking issues in its algorithmic framework?*

As mentioned in (Dong et al., 2020, Section 2.3), Corollary C.2 in Madry et al. (2018) can be considered the theoretical optimization foundation of the non-convex non-concave min-max optimization algorithms that we now collectively refer to as *Adversarial Training*. It justifies the two-stage structure of the training loop: first we find one approximately optimal adversarial example and then we update the model using the gradient (with respect to the model parameters) at the perturbed input.

The only drawbacks of a first-order adversary seem to be its computational complexity and its approximate suboptimal solver nature. Ignoring the computational complexity issue, suppose we have access to a theoretical oracle that provides *a single solution* of the inner-maximization problem. *In such idealized setting, can we safely assume AT is decreasing the adversarially robust loss on the data sample?* According to the aforementioned theoretical results, it would appear so.

In this work, we scrutinize the optimization paradigm on which Adversarial Training (AT) has been founded, and we posit that finding multiple solutions of the inner-maximization problem is necessary

---

[*]These authors contributed equally to this work

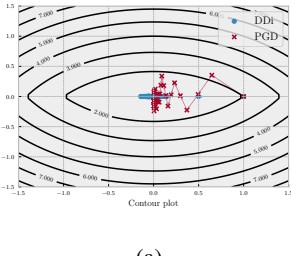 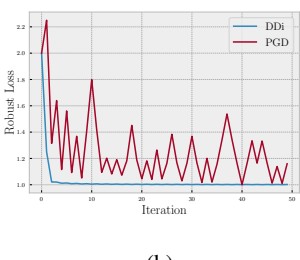 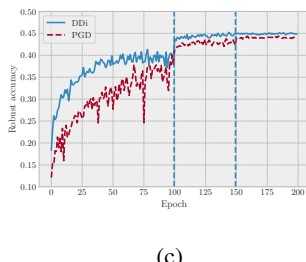

|  |  |  |
| :---: | :---: | :---: |
| (a) | (b) | (c) |

Figure 1: (a) and (b): comparison of our method (DDi) and the single-adversarial-example method (PGD) on a synthetic min-max problem. Using a single example may increase the robust loss. DDi computes 10 examples and can avoid this. (c): similar improvement over PGD training shown on CIFAR10, where DDi with 10 examples speeds up convergence. More details in Section 5

to find good descent directions of the adversarially robust loss. In doing so, we hope to improve our understanding of the non-convex/non-concave min-max optimization problem that underlies the Adversarial Training methodology, and potentially improving its performance.

**Our contributions:** We present two counterexamples to Madry et al. (2018, Corollary C.2), the motivation behind AT. They show that using the gradient (with respect to the parameters of the model) evaluated at a *single* solution of the inner-maximization problem, can increase the robust loss, i.e., it can harm the robustness of the model. In particular, in counterexample 2 many descent directions exist, but they cannot be found if we only compute a single solution of the inner-maximization problem. In Section 2 we explain that the flaw in the proof is due to a misunderstanding of the directional derivative notion that is used in the original work of Danskin (1966).

Based on our findings, we propose *Danskin's Descent Direction* (DDi, Algorithm 1). It aims to overcome the problems of the *single adversarial example* paradigm of AT by exploiting multiple adversarial examples, obtaining better update directions for the network. For a data-label pair, DDi finds the *steepest descent direction* for the robust loss, assuming that ($i$) there exists a finite number of solutions of the inner-maximization problem and ($ii$) they can be found with first-order methods.

In Section 5 we verify experimentally that: ($i$) it is unrealistic to assume a unique solution of the inner-maximization problem, hence making a case for our method DDi, ($ii$) our method can achieve more stable descent dynamics than the vanilla AT method in synthetic scenarios and ($iii$) on the CIFAR10 dataset DDi is more stable and achieves higher robustness levels in the early stages of traning, compared with a PGD adversary of equivalent complexity. This is observed in a setting where the conditions of Danskin's Theorem holds, i.e., using differentiable activation functions and removing BatchNorm. As a limitation, PGD training of ReLU+BatchNorm networks still performs better, but there is no theory explaining this. The code to reproduce our results will be available at `https://github.com/LIONS-EPFL/ddi_at`.

**Remark.** The fact that (Madry et al., 2018, Corollary C.2) is false, might be well-known in the optimization field. In the convex setting it corresponds to the common knowledge that a negative subgradient of a non-smooth convex function might not be a descent direction c.f., (Boyd, 2014, Section 2.1). However, we believe this is not well-known in the AT community given that ($i$) its practical implications i.e., methods deriving steeper descent updates using multiple adversarial examples, have not been previously introduced, and ($ii$) the results in Madry et al. (2018) have been central in the development of AT. Hence, our contribution can be understood as raising awareness about the issue, and demonstrating its practical implications for AT.

## 2 A COUNTEREXAMPLE TO MADRY ET AL. (2018, COROLLARY C.2)

**Preliminaries.** Let $\theta \in \mathbb{R}^d$ be the parameters of a model, $(x, y) \sim \mathcal{D}$ a data-label distribution, $\delta$ a perturbation in a compact set $\mathcal{S}_0$ and $L$ a loss function. The optimization objective of AT is:

$$\min_\theta \rho(\theta), \qquad \text{where } \rho(\theta) := \mathbb{E}_{(x,y)\sim\mathcal{D}} \left[ \max_{\delta \in \mathcal{S}_0} L(\theta, x + \delta, y) \right] \tag{1}$$

In this setting $\rho(\theta)$ is referred to as the *adversarial loss* or *robust loss*. In order to optimize Eq. (1) via iterative first-order methods, we need access to an stochastic gradient of the adversarial loss $\rho$ or at least, the weaker notion of *stochastic descent direction* i.e., a direction along which the function

$$\phi(\theta) := \max_{\delta \in \mathcal{S} := \mathcal{S}_0^k} \left\{ g(\theta, \delta) := \frac{1}{k} \sum_{i=1}^{k} L(\theta, x_i + \delta_i, y_i) \right\} \qquad (2)$$

decreases in value. We have collected the perturbations $\delta_i \in \mathcal{S}_0$ on the batch $\{(x_i, y_i)\}_{i=1}^k$ as the columns of a matrix $\delta = [\delta_1, \ldots, \delta_k] \in \mathcal{S} := \mathcal{S}_0^k$ which is also a compact set. To obtain a descent direction for *partial maximization* functions like $\phi$ we resort to Danskin's Theorem:

**Theorem 1** (Danskin (1966)). *Let $\mathcal{S}$ be a compact topological space, and let $g : \mathbb{R}^d \times \mathcal{S}$ be a continuous function such that $g(\cdot, \delta)$ is differentiable for all $\delta \in \mathcal{S}$ and $\nabla_\theta g(\theta, \delta)$ is continuous on $\mathbb{R}^d \times \mathcal{S}$. Let*

$$\phi(\theta) := \max_{\delta \in \mathcal{S}} g(\theta, \delta), \qquad \mathcal{S}^\star(\theta) := \arg\max_{\delta \in \mathcal{S}} g(\theta, \delta) \qquad (3)$$

*Let $\gamma \in \mathbb{R}^d$ with $\|\gamma\|_2 = 1$ be an arbitrary unit vector. The directional derivative $D_\gamma \phi(\theta)$ of $\phi$ in the direction $\gamma$ at the point $\theta$ exists, and is given by the formula*

$$D_\gamma \phi(\theta) = \max_{\delta \in \mathcal{S}^\star(\theta)} \langle \gamma, \nabla_\theta g(\theta, \delta) \rangle \qquad (4)$$

**Remark.** $\gamma \neq 0$ is called a descent direction of $\phi$ at $\theta$ if and only if $D_\gamma \phi(\theta) < 0$, i.e., if the directional derivative is strictly negative.

Corollary 1 is an equivalent rephrasing of Madry et al. (2018, Corollary C.2.), and was originally claimed to be a consequence of Theorem 1. Unfortunately counterexample 1 shows that the corollary is false. As Theorem 1 (Danskin's Theorem) is true, this means that there is some mistake in the proof of the corollary provided in Madry et al. (2018).

**Corollary 1.** *Let $\delta^\star \in \mathcal{S}^\star(\theta)$. If $-\nabla_\theta g(\theta, \delta^\star) \neq 0$, then it is a descent direction for $\phi$ at $\theta$.*

**Counterexample 1.** *Let $\mathcal{S} := [-1, 1]$ and $g(\theta, \delta) = \theta\delta$. The conditions of Danskin's theorem clearly hold in this case, and*

$$\phi(\theta) := \max_{\delta \in [-1,1]} \theta\delta = |\theta|. \qquad (5)$$

*Note that at $\theta = 0$, we have $\mathcal{S}^\star(0) = [-1, 1]$. Choosing $\delta = 1 \in \mathcal{S}^\star(0)$ we have that $g(\theta, 1) = \theta$ and so $-\nabla_\theta g(0, 1) = -1 \neq 0$. Hence, Corollary 1 would imply that $-1$ is a descent direction for $\phi(\theta) = |\theta|$. However, $\theta = 0$ is a global minimizer of the absolute value function, which means that there exists no descent direction. This is a contradiction.*

To cast more clarity on why Corollary 1 is false, we explain what is the mistake in the proof provided in Madry et al. (2018). The main issue is the definition of the *directional derivative*, a concept in multivariable calculus that is defined in slightly different ways in the literature.

**Definition 1.** *Let $\phi : \mathbb{R}^d \to \mathbb{R}$. For a nonzero vector $\gamma \in \mathbb{R}^d$, the one-sided directional derivative of $\phi$ in the direction $\gamma$ at the point $\theta$ is defined as the one-sided limit:*

$$D_\gamma \phi(\theta) := \lim_{t \to 0^+} \frac{\phi(\theta + t\gamma) - \phi(\theta)}{t\|\gamma\|_2} \qquad (6)$$

*The two-sided directional derivative is defined as the two-sided limit:*

$$\hat{D}_\gamma \phi(\theta) := \lim_{t \to 0} \frac{\phi(\theta + t\gamma) - \phi(\theta)}{t\|\gamma\|_2} \qquad (7)$$

Unfortunately, it is not always clear which one of the two notions is meant when the term *directional derivative* is used. Indeed, as our notation suggests, the one-sided definition Eq. (6) is the one used in the statement of Danskin's Theorem (Danskin, 1966). However, the proof of Corollary 1 provided in Madry et al. (2018) mistakenly assumes the two-sided definition Eq. (7), and inadvertently uses the following property that holds for $\hat{D}_\gamma \phi(\theta)$ (Eq. (7)) but not for $D_\gamma \phi(\theta)$ (Eq. (6)):

**Lemma 1.** *For the two-sided directional derivative definition (7) it holds that $-\hat{D}_\gamma \phi(\theta) = \hat{D}_{-\gamma} \phi(\theta)$ provided that $\hat{D}_\gamma$ exists. In particular, if $\hat{D}_\gamma \phi(\theta) > 0$ then $\hat{D}_{-\gamma} \phi(\theta) < 0$. However this is not true for the one-sided directional derivative (6), as the example $\phi(\theta) = |\theta|$ at $\theta = 0$ shows (both directional derivatives are strictly positive).*

We provide a proof of this fact in Appendix E. The (flawed) proof of Corollary 1 provided in Madry et al. (2018) starts by noting that for a solution $\bar{\delta}$ of the inner-maximization problem, the directional derivative in the direction $\gamma = \nabla_\theta g(\theta, \bar{\delta})$ is positive, as implied by Danskin's Theorem:

$$D_\gamma \phi(\theta) = \max_{\delta \in \mathcal{S}^\star(\theta)} \langle \gamma, \nabla_\theta g(\theta, \delta) \rangle \geq \langle \nabla_\theta g(\theta, \bar{\delta}), \nabla_\theta g(\theta, \bar{\delta}) \rangle = \|\nabla_\theta g(\theta, \bar{\delta})\|^2 > 0 \qquad (8)$$

assuming that $\nabla_\theta g(\theta, \bar{\delta})$ is non-zero. The mistake in the proof lies in concluding that $D_{-\gamma}\phi(\theta) < 0$. Following Lemma 1, this property does not hold for the one-sided directional derivative definition Eq. (6), the one used in Danskin's Theorem.

## 3  A COUNTEREXAMPLE AT A POINT THAT IS NOT LOCALLY OPTIMAL

The question remains whether a slightly modified version of Corollary 1 holds true: it might be the case that by adding some mild assumption, we exclude all possible counterexamples. In the particular case of counterexample 1, $\theta = 0$ is a local optimum of the function $\phi(\theta) = |\theta|$. At such points, descent directions do not exist. However, in the trajectory of an iterative optimization algorithm we are mostly concerned with non-locally-optimal points. Hence, we explore whether adding the assumption that $\theta$ is not locally optimal can make Corollary 1 true. Unfortunately, we will show that this is not the case.

To this end we construct a family of counterexamples to Corollary 1 with the following properties: $(i)$ there exists a descent direction at a point $\theta$ (that is, $\theta$ is not locally optimal) and $(ii)$, it does not coincide with $-\nabla_\theta g(\theta, \delta)$, for any optimal $\delta \in \mathcal{S}^\star(\theta)$. Moreover, all the directions $-\nabla_\theta g(\theta, \delta)$ are in fact *ascent directions* i.e., they lead to an increase in the function $\phi(\theta)$.

**Counterexample 2.** *Let $\mathcal{S} := [0, 1]$ and let $u, v \in \mathbb{R}^2$ be unit vectors such that $-1 < \langle u, v \rangle < 0$. That is, $u$ and $v$ form an obtuse angle. Let*

$$g(\theta, \delta) = \delta \langle \theta, u \rangle + (1 - \delta) \langle \theta, v \rangle + \delta(\delta - 1) \qquad (9)$$

*Clearly, the function satisfies all conditions of Theorem 1. At $\theta = 0$, we have that $\mathcal{S}^\star(0) = \arg\max_{\delta \in [0,1]} \delta(\delta - 1) = \{0, 1\}$. At $\delta = 0$ we have $\nabla_\theta g(\theta, 0) = \nabla_\theta \langle \theta, v \rangle = v$ and at $\delta = 1$ we have $\nabla_\theta g(\theta, 1) = \nabla_\theta \langle \theta, u \rangle = u$. We compute the value of the directional derivatives in the negative direction of such vectors. According to Danskin's Theorem we have*

$$D_{-v}\phi(0) = \max_{\delta \in \{0,1\}} \langle -v, \nabla_\theta g(\theta, \delta) \rangle = \max(\langle -v, v \rangle, \langle -v, u \rangle) \geq -\langle v, u \rangle > 0 \qquad (10)$$

*where $-\langle v, u \rangle > 0$ holds by construction. Analogously, $D_{-u}\phi(0) > 0$. This means that all such directions are ascent directions. However, for the direction $\gamma = -(u + v)$ we have*

$$D_\gamma \phi(\theta) \propto \max_{\delta \in \{0,1\}} \langle -(u + v), \nabla_\theta g(\theta, \delta) \rangle$$
$$= \max(\langle -u - v, u \rangle, \langle -u - v, v \rangle) = -1 - \langle u, v \rangle < 0 \qquad (11)$$

*where the last inequality also follows by construction. Hence, $-(u + v)$ is a descent direction.*

As counterexample 2 shows, Adversarial Training has the following problem: even if we are able to compute one solution of the inner-maximization problem $\bar{\delta} \in \mathcal{S}$ it can be the case that moving in the direction $-\nabla_\theta g(\theta, \bar{\delta})$ increases the robust training loss i.e., the classifier becomes less, rather than more, robust. This can happen at any stage, independently of the local optimality of $\theta$.

For a non-locally-optimal $\theta \in \mathbb{R}^d$, the construction of the counterexamples relies on the following: if for any gradient computed at one inner-max solution, there exist another gradient (at a different inner-max solution) forming an obtuse angle, then no single inner-max solution yields a descent direction. Consequently, it suffices to ensure that for any gradient that can be found by solving the inner problem, there exists another one that has a negative inner product with it. Precisely, our counterexample 2 is carefully crafted so that this property holds.

## 4  DANSKIN'S DESCENT DIRECTION

Danskin's Theorem implies that the directional derivative depends on *all* the solutions of the inner-max problem $\mathcal{S}^\star(\theta)$ c.f., Eq. (4). One possible issue in Adversarial Training is relying on a single

solution, as it does not necessarily lead to a descent direction c.f. counterexample 2. To fix this, we design an algorithm that uses multiple adversarial perturbations per data sample. In theory, we can obtain the *steepest descent direction* for the robust loss on a batch $\{(x_i, y_i) : i = 1, \ldots, k\}$ by solving the following min-max problem:

$$\gamma^\star \in \arg\min_{\gamma : \|\gamma\|_2 = 1} \max_{\delta \in \mathcal{S}^\star(\theta)} \langle \gamma, \nabla_\theta g(\theta, \delta) \rangle, \qquad g(\theta, \delta) := \frac{1}{k} \sum_{i=1}^k L(\theta, x_i + \delta_i, y_i) \qquad (12)$$

On the one hand, if the set of maximizers $\mathcal{S}^\star(\theta)$ is infinite, Eq. (12) would be out of reach for computationally tractable methods. On the other hand, the solution is trivial if there is a single maximizer , but we verify experimentally in Section 5 that such assumption is wrong in practice. In conclusion, a compromise has to be made in order to devise an tractable algorithm that is relevant in practical scenarios. First, we assume that the set of optimal adversarial perturbations is finite:

$$\mathcal{S}^\star(\theta) := \arg\max_{\delta \in \mathcal{S}} g(\theta, \delta) = \mathcal{S}_m^\star(\theta) = \{\delta^{(1)}, \ldots, \delta^{(m)}\}, \qquad m \geq 1, m \in \mathbb{Z} \qquad (13)$$

Under such assumption, it is possible to compute the steepest descent direction in Eq. (12) efficiently.

**Theorem 2.** *Let $\Delta^m$ be the $m$-dimensional simplex i.e., $\alpha \geq 0, \sum_{i=1}^m \alpha_i = 1$. Suppose that $\mathcal{S}^\star(\theta) = \mathcal{S}_m^\star(\theta) := \{\delta^{(1)}, \ldots, \delta^{(m)}\}$ and denote by $\nabla_\theta g(\theta, \mathcal{S}_m^\star(\theta))$ the matrix with columns $\nabla_\theta g(\theta, \delta^{(i)})$ for $i = 1, \ldots, m$. As long as $\theta$ is not a local minimizer of the robust loss $\phi(\theta) = \max_{\delta \in \mathcal{S}} g(\theta, \delta)$, then the steepest descent direction of $\phi$ at $\theta$ can be computed as:*

$$\gamma^\star := -\frac{\nabla_\theta g(\theta, \mathcal{S}_m^\star(\theta))\alpha^\star}{\|\nabla_\theta g(\theta, \mathcal{S}_m^\star(\theta))\alpha^\star\|}, \qquad \alpha^\star \in \arg\min_{\alpha \in \Delta^m} \|\nabla_\theta g(\theta, \mathcal{S}_m^\star(\theta))\alpha\|_2^2 \qquad (14)$$

We present the proof of Theorem 2 in Appendix C. We now relax our initial finiteness assumption Eq. (13), as it might not hold in practice. We show that it might suffice to *approximate* the (possibly infinite) set of maximizers $\mathcal{S}^\star(\theta)$ with a finite set $\mathcal{S}_m^\star(\theta)$. If the direction $\gamma^\star$ defined in Eq. (14) satisfies an additional inequality involving the finite set $\mathcal{S}_m^\star(\theta)$, it will be a certified descent direction.

**Theorem 3.** *Suppose that $\nabla_\theta g(\theta, \delta)$ is $L$-Lipschitz as a function of $\delta$, i.e., $\|\nabla_\theta g(\theta, \delta) - \nabla_\theta g(\theta, \delta')\|_2 \leq L\|\delta - \delta'\|_2$. Let $\mathcal{S}^\star(\theta)$ be the set of solutions of the inner maximization problem, and let $\mathcal{S}_m^\star(\theta) := \{\delta^{(1)}, \ldots, \delta^{(m)}\}$ be a finite set that $\epsilon$-approximates $\mathcal{S}^\star(\theta)$ in the following sense: for any $\delta \in \mathcal{S}^\star(\theta)$ there exists $\delta^{(i)} \in \mathcal{S}_m^\star(\theta)$ such that $\|\delta - \delta^{(i)}\|_2 \leq \epsilon$. Let $\gamma^\star$ be as in Eq. (14). If $\max_{\delta \in \mathcal{S}_m^\star(\theta)} \langle \gamma^\star, \nabla_\theta g(\theta, \delta) \rangle < -L\epsilon$ then $\gamma^\star$ is a descent direction for $\phi$ at $\theta$.*

The Lipschitz gradient assumption in Theorem 3 is standard in the optimization literature. We provide a proof of Theorem 3 in Appendix D. This results motivate *Danskin's Descent Direction* (Algorithm 1). We assume an oracle providing a finite set of adversarial perturbations $\mathcal{S}_m^\star(\theta)$ that satifies the approximation assumption in Theorem 3. In particular, this does not require solving the inner-maximization problem to optimality, which is out of reach for computationally tractable methods and requires expensive branch-and-bound or MIP techniques (Zhang et al., 2022; Tjeng et al., 2019; Palma et al., 2021; Wang et al., 2021). Given $\mathcal{S}_m^\star(\theta)$, we compute $\gamma^\star$ as in Eq. (14), which corresponds to Line 7 of Algorithm 1. If the values of $L$ and $\epsilon$ in Theorem 3 are not available (they might be hard to compute), we cannot certify that $\gamma^\star$ is a descent direction. However, note that given a set of adversarial examples $\mathcal{S}_m^\star(\theta)$, $\gamma^\star$ is still the best choice as it ensures we improve the loss on all elements of $\mathcal{S}_m^\star(\theta)$.

The optimization problem defining $\alpha^\star$ and $\gamma^\star$ can be solved to arbitrary accuracy efficiently: It corresponds to the minimization of a smooth objective subject to the convex constraint $\alpha \in \Delta^m$. We use the accelerated PGD algorithm proposed in (Parikh et al., 2014, section 4.3) and pair it with the efficient simplex projection algorithm given in Duchi et al. (2008). As the problem is smooth, a fixed step-size choice guarantees convergence. We set it as the inverse of the spectral norm of $\nabla_\theta g(\theta, \mathcal{S}^\star(\theta))^\top \nabla_\theta g(\theta, \mathcal{S}^\star(\theta))$ and run the algorithm for a fixed number of iterations. Alternatively, one can consider Frank-Wolfe with away steps (Lacoste-Julien & Jaggi, 2015).

In practice, the theoretical oracle algorithm that computes the set $\mathcal{S}_m^\star(\theta)$ is replaced by heuristics like performing multiple runs of the Fast Gradient Sign Method (FGSM) or Iterative FGSM (Kurakin et al., 2017) (referred to as PGD in Madry et al. (2018)). The complexity of an iteration in Algorithm 1 depends on this choice. In Section 5 we explore different choices and how it affects the the performance of the method.

---

**Algorithm 1** Danskin's Descent Direction (DDi)

---

1: **Input:** Batch size $k \geq 1$, number of adversarial examples $m$, initial iterate $\theta_0 \in \mathbb{R}^d$, number of iterations $T \geq 1$, step-sizes $\{\beta_t\}_{t=1}^T$.
2: **for** $t = 0$ **to** $T - 1$ **do**
3:      Draw $(x_1, y_1), \ldots, (x_k, y_k)$ from data distribution $\mathcal{D}$
4:      $g(\theta, \delta) \leftarrow \frac{1}{k} \sum_{i=1}^k L(\theta, x + \delta_i, y_i)$
5:      $\delta^{(1)}, \ldots, \delta^{(m)} \leftarrow \text{MAXIMIZE}_{\delta \in \mathcal{S}} g(\theta_t, \delta)$            ▷ Using a heuristic like PGD
6:      $M \leftarrow \left[ \nabla_\theta g(\theta_t, \delta^{(i)}) : i = 1, \ldots, m \right] \in \mathbb{R}^{d \times m}$
7:      $\alpha^\star \leftarrow \text{MINIMIZE}_{\alpha \in \Delta^m} \|M\alpha\|_2^2$            ▷ To $\epsilon$-suboptimality
8:      $\gamma^\star \leftarrow \frac{M\alpha^\star}{\|M\alpha^\star\|_2}$
9:      $\theta_{t+1} \leftarrow \theta_t + \beta_t \gamma^\star$
10: **end for**
11: **return** $\theta_T$

---

## 5 EXPERIMENTS

### 5.1 EXISTENCE OF MULTIPLE OPTIMAL ADVERSARIAL SOLUTIONS

This section provides evidence that the set of optimal adversarial examples for a given sample is not a singleton. The hypothesis is tested by using a ResNet-18 pretrained on CIFAR10 and computing multiple randomly initialized PGD-7 attacks for each image with $\varepsilon = \frac{8}{255}$. We compute all pairwise $\ell_2$-distances between attacks for a given image and plot a joint histogram for 10 examples in Figure 2. There is a clear separation away from zero for all pairwise distances indicating that the attacks are indeed distinct in the input space. Additionally, we plot a histogram over the adversarial losses for each image. An example is provided in Figure 2, which is corroborated by similar results for other images (see Figure 6§B). We find that the adversarial losses all concentrate with low variance far away from the clean loss. This confirms that all perturbations are in fact both strong and distinct.

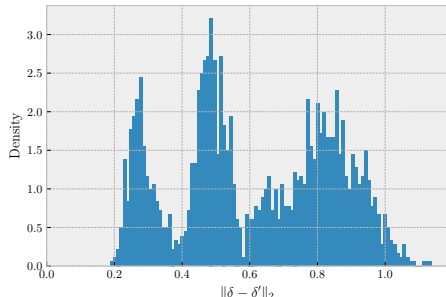 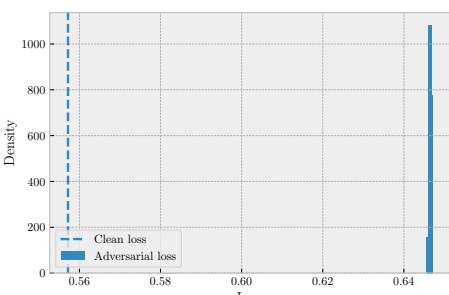

Figure 2: Non-uniqueness of an optimal adversarial perturbation. (left) Pairwise $\ell_2$-distances between PGD-based perturbations are bounded away from zero by a large margin, showing that they are distinct. (right) The losses of multiple perturbations on the same sample concentrate around a value much larger than the clean loss (see Fig. 7 for zoomed-in version).

### 5.2 EXPLORING THE OPTIMIZATION LANDSCAPE OF DDI AND STANDARD ADVERSARIAL TRAINING

Having established that there exist multipe adversarial examples, we now show that the gradients computed can exhibit the behaviors discussed in Section 3. In a first synthetic example we borrow from (Orabona, 2019, Chapter 6), we consider the function $g(\theta, \delta) = \delta \left( \theta_1^2 + (\theta_2 + 1)^2 \right) + (1 - \delta) \left( \theta_1^2 + (\theta_2 - 1)^2 \right)$ where $\theta \in \mathbb{R}^2$ and $\delta \in [0, 1]$. As can be seen from Figure 1a and Figure 1b, following a gradient computed at a single example leads to a increase in the objective and an unstable optimization behavior despite the use of a decaying step-size.

In a second synthetic examples, we consider robust binary classification with a feed-forward neural network on a synthetic 2-dimensional dataset, trained with batch gradient descent. We observe that during training, after an initial phase where all gradients computed at different perturbations point roughly in the same direction, we begin to observe pairs of gradients with negative inner-products (see Figure 3 (left)). That means that following one of those gradients would lead to an increase of the robust loss, as shown by the different optimization behavior (see Figure 3 (center)). Therefore, the benefits DDi kick in later in training, once the loss has stabilized and the inner-solver starts outputting gradients with negative inner products. Indeed, we see that in the middle of training (iteration 250), DDi finds a descent direction of the (linearized) robust objective, whereas all individual gradients lead to an increase.

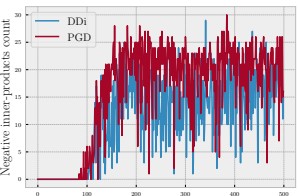 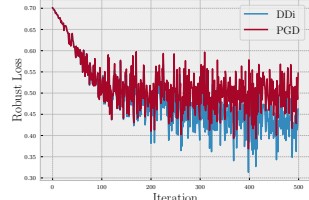 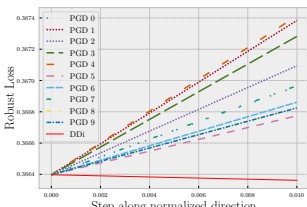

Figure 3: Count of negative inner products pairs among the 10 gradients computed per iteration(left), corresponding robust loss behavior along optimization (center). At iteration 250, comparison of the direction obtained by DDi and individual gradients.(right).

### 5.3 ACCURACY/ROBUSTNESS COMPARISON OF DDI VS ADVERSARIAL TRAINING

We compare the robust test and training error of Adversarial Training vs our proposed method DDi, on the CIFAR10 benchmark. As baseline we use $\ell_\infty$-PGD with $\epsilon = 8/255, \alpha = 2/255, n_{inner} = 7$. We train a ResNet18 with SGD, using the settings from Pang et al. (2021), Table 1 except for some modifications noted below. This means SGD with hyperparameters `lr`= 0.1, `momentum`=0.0 (not the default 0.9, we explain why below), `batch_size`= 128 and `weight_decay`= $5e - 4$. We run for 200 epochs, no warmup, decreasing `lr` by a factor of 0.1 at 50% and 75% of the epochs.

**Satisfying theoretical assumptions:** Real world architectures are often not covered by theory while simple toy examples are often far removed from practice. To demonstrate the real world impact of our results, we therefore study a setting where the conditions of Danskin's Theorem hold, but which also uses standard building blocks used by practitioners, specifically replacing ReLU with CELU(Barron, 2017), replacing BatchNorm (BN) (Ioffe & Szegedy, 2015) with GroupNorm (GN) (Wu & He, 2018) and removing momentum. This ensures differentiability, removes intra-batch dependencies and ensures each update depends only on the descent direction found at that step respecively. We present more detailed justification in Appendix B.2 due to space constraints and additionally show an ablation study on the effect of our modifications in (Section 5.3) [1].

Our main results can be seen in Section 5.3. The robust accuracy of the DDi-trained model increases much more rapidly in the early stages, it increases more after the first drop in the learning rate, and is more stable when compared to the baseline. Section 5.3 also gives evidence that our method has (generally positive or neutral) effects in all settings. Using $ReLU$ instead of $CELU$ re-introduces the characteristic *bump* in robust accuracy that has led to early stopping becoming standard practice in robust training. It also diminishes the benefit of DDi, but DDi remains on par with PGD in terms of training speed and decays slightly less towards the end of the training. Adding momentum does not help either method in terms of training speed and makes them behave almost identically.

Finally, BN seems to significantly ease the optimisation for both methods, raising overall performance and amplifying the *bump* on both methods. Here, PGD actually reaches a higher maximum robust accuracy and rises faster initially, but then converges to a lower value. This implies that some benefits of DDi remain even outside the setting covered by the theory.

---

[1]It is worth noting that the early stopping robust accuracy we achieve in ablations approximately matches that reported in Engstrom et al. (2019) on resnet50

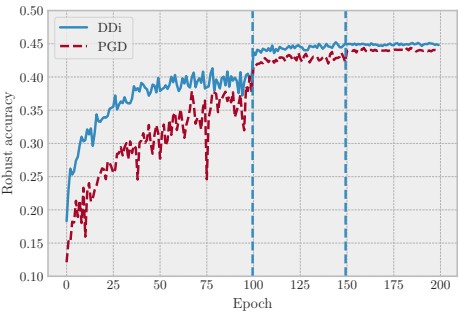 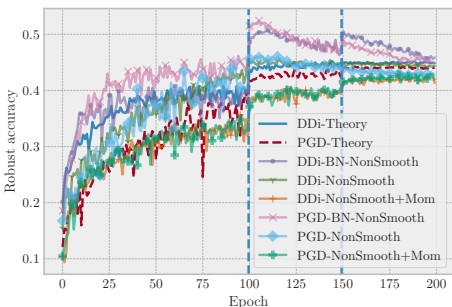

Figure 4: (left) Evolution of the robust accuracy on the CIFAR10 validation set, using a standard PGD-20 adversary for evaluation and DDi/PGD-7 during training. (right) an ablation testing the effect of adding the elements not covered by theory (BN,ReLU,momentum) back into our setting.

Although these are promising results indicating that DDi can give real world benefits in terms of iterations and reduce the need for early stopping, it is worth asking whether once could get the same benefit with a simpler or cheaper method. The final robust accuracies obtained are very close, and the increased convergence rate in terms of *steps* comes at a more than 10x slowdown due to having to perform 10 independent forward-backward passes and then solving an additional inner problem. Additionally, it could be argued that these results are to be expected and trivial: we are spending 10x the compute to get 10x the gradients.

One might even say there is no need to solve the inner product and a simpler method to select the *best* adversary would suffice. In Fig. 5a we address these concerns by comparing Section 5.3 to the results of the following variants attempting to match the computational complexity: *PGD*-70 runs a single PGD adversary for 10x the number of steps, *PGD*-70 $- \frac{1}{t}$ runs a single PGD adversary for 10x the number of steps, using a $1/t$ learning rate decay after leaving the "standard" PGD regime (i.e. after 8 adversary steps) to converge closer to an optimal adversarial example, *PGD-max-10* runs *ten* parallel, independent PGD adversaries for each image and select the adversarial example that induces the largest loss. Finally, *PGD-min-10* runs *ten* parallel, independent PGD adversaries for each image, *then computes the gradients* and selects the one with the lowest norm.This is an approximation of DDi that avoids solving Line 7 in Algorithm 1.

In Fig. 5b we create a DDi variant based on the FAST adversary (Wong et al., 2020) (using $\epsilon = 8/255, \alpha = 10/255$). Using PGD for the evaluation attack, we compare against vanilla FAST in our setting (no BN, momentum and using CELU) as well as a FAST-max-10 variant analoguous to PGD-max-10. As we can see in Fig. 5a, every step of the pipeline of DDi seems to be necessary, with none of the PGD variants achieving the fast initial rise in robustness. PGD-70 $- \frac{1}{t}$ and PGD-min-10 reach a higher final robust accuracy, which we attribute to the higher quality adversarial example and informed selection respectively. This is corroborated in Fig. 5b. Using a single step adversary is sufficient to speed up convergence in the early stages of training, but does not reach the same final robust accuracy.

PGD and DDi seem to behave similarly in the later stages of training. We would suggest a computationally cheaper DDi variant which uses single ascent steps (FAST) in the beginning of training and PGD in the later stages. In any case, the bulk of the overhead lies in the subroutine in Line 7 of Algorithm 1. A faster approximate solution could also speed up the method significantly. Such incremental improvements are left for future work Neverthelss, in Appendix B.4 we explore some modifications that can reduce the runtime of Algorithm 1 by at least 70% while retaining its benefits.

## 6 RELATED WORK

Wang et al. (2019) derive suboptimality bounds for the robust training problem, under a locally strong concavity assumption on the inner-maximization problem. However, such results do not extend to Neural Networks, as the inner-maximization problem is not strongly concave, in general.

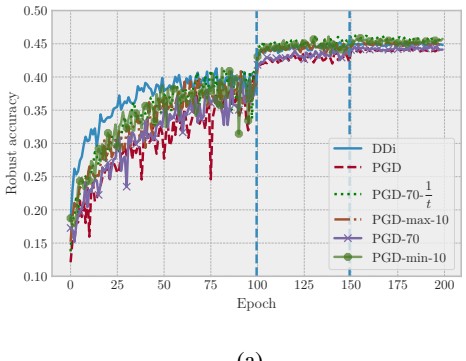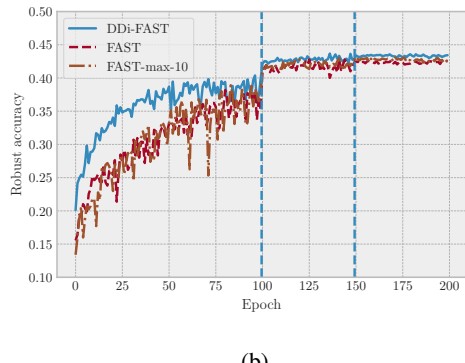

(a)                                         (b)

Figure 5: (a) Ablations comparing PGD-variants matching the number of adversarial gradients/steps used for DDi. (b) Ablation over single-step adversaries (FAST/DDi-FAST).

In contrast, we do not make unrealistic assumptions like strong concavity, and we deal with the existence multiple solutions of the inner-maximization problem.

In Nouiehed et al. (2019), it is shown that if the inner-maximization problem is unconstrained and satisfies the PL-condition, it is differentiable, and the gradient can be computed after obtaining a single solution of the problem. However, in the robust learning problem the adversary is usually constrained to a compact set, and the PL condition does not hold generically. This renders such assumptions hard to justify in the AT setting.

Tramer & Boneh (2019); Maini et al. (2020) study robustness to multiple *perturbation types*, which might appear similar to our approach, but is not. Such works strike to train models that are simultaneously robust against $\ell_\infty$- and $\ell_2$-bounded perturbations, for example. In contrast, we focus on a single perturbation type, and we study how to use multiple adversarial examples of the same sample to improve the update directions of the network parameters.

Finally, we back our claim that the falseness of Madry et al. (2018, Corollary C.2.) is not well-known in the literature on Adversarial Training. For example, such result is included in the textbook (Vorobeychik et al., 2018, Proposition 8.1). It has also been either reproduced or mentioned in conference papers like Liu et al. (2020, Section 2), Viallard et al. (2021, Appendix B), Wei & Ma (2020, Section 5) and possibly many others. This supports our claim that raising awareness about the mistake in the proof is an important contribution.

## 7    CONCLUSION

In this paper we presented a formal proof, counter examples and evidence about the real world impact of the fact that a foundational corollary of the Adversarial Training literature is in fact false. Raising awareness about an incorrect claim that has been present in the Adversarial Training literature may provide opportunities to develop improved variants of the method. Indeed, we see some improvents in an implementable algorithm that align with our theoretical arguments: DDi exploits multiple approximate solutions of the inner-maximization problem, yields better updates for the parameters of the network and improves the optimization dynamics. However, it is important to remember the limitations and opportunities for future work: our algorithm requires multiple forward-backward passes and one additional optimization problem. Reducing the overhead over the vanlla PGD method would certainly make our results truly practical.

Non-smooth activations and the use of Batch Normalization or momentum still falls outside the scope of existing theory but might achieve better performance in benchmarks. To date, this requires using precise hyperparameters and tricks like early-stopping, that have only been found to work a-posteriori through extensive trial and error. Since we observe lower decay even in such setting, future work extending the analysis to cover this case might help alleviate this cost.

## ACKNOWLEDGMENTS

This work is funded (in part) through a PhD fellowship of the Swiss Data Science Center, a joint venture between EPFL and ETH Zurich. Igor Krawczuk, Leello Dadi, Thomas Pethick and Volkan Cevher acknowledge funding from the European Research Council (ERC) under the European Union's Horizon 2020 research and innovation programme (grant agreement n° 725594 - time-data). This work was supported by the Swiss National Science Foundation (SNSF) under grant number 200021_205011.

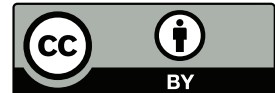

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

## A MORE ON COUNTEREXAMPLES

Here we give more details on the construction of the counterexamples. First observe that for a given point $\theta_0$, and a direction $\gamma$, if there exists a $\delta_0 \in \mathcal{S}^\star(\theta_0)$ such that $\langle \gamma, \nabla_\theta g(\theta_0, \delta) \rangle > 0$, then $\gamma$ is not a descent direction since $D_\gamma \phi(\theta_0) \geq 0$.

In order to ensure that no descent directions can be recovered by solving the inner-maximization, it suffices to guarantee that for any $\delta \in \mathcal{S}^\star(\theta_0)$, there exists $\delta' \in \mathcal{S}^\star(\theta_0)$ such that $\langle \nabla_\theta g(\theta_0, \delta'), \nabla_\theta g(\theta_0, \delta) \rangle < 0$. This way, neither $-\nabla_\theta g(\theta_0, \delta)$ nor $-\nabla_\theta g(\theta_0, \delta')$ would be descent directions.

It easy to generate instances verifying the above using linear functions. More formally, by taking any family of vectors $\mathcal{V} = \{v_1, \ldots, v_n\}$ such that for any $i \in \{1, \ldots, n\}$ there exists $j \in \{1, \ldots, n\}$ such that $\langle v_i, v_j \rangle < 0$, we can construct the objective $g(\theta, \delta) = \sum \delta_i v_i^\top (\theta - \theta_0) - H(\delta)$, where $\delta$ is in the $n$-dimensional Simplex and $H$ is the Shannon entropy. Solving the inner-maximization would yield any one of the vectors $\{v_1, \ldots, v_n\}$, and by construction, none of them are descent directions.

## B EXPERIMENTS

### B.1 MULTIPLE ATTACKS

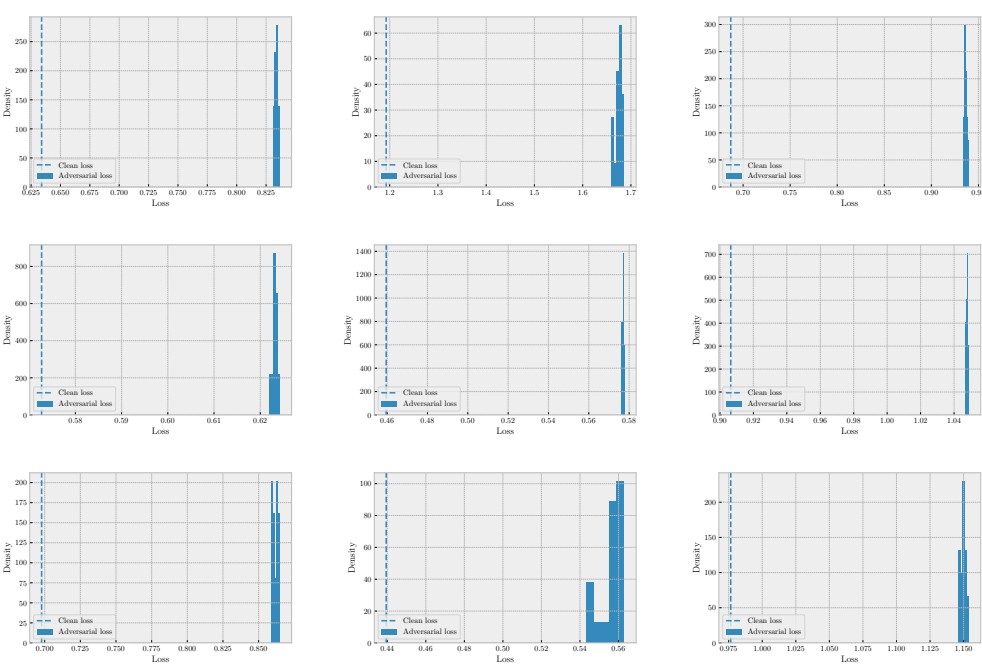

Figure 6: The losses of multiple perturbations on 9 different example all concentrate around a value much larger than the clean loss. See Section 5.1 for experimental details. The histograms have been enlarged in Figure 7.

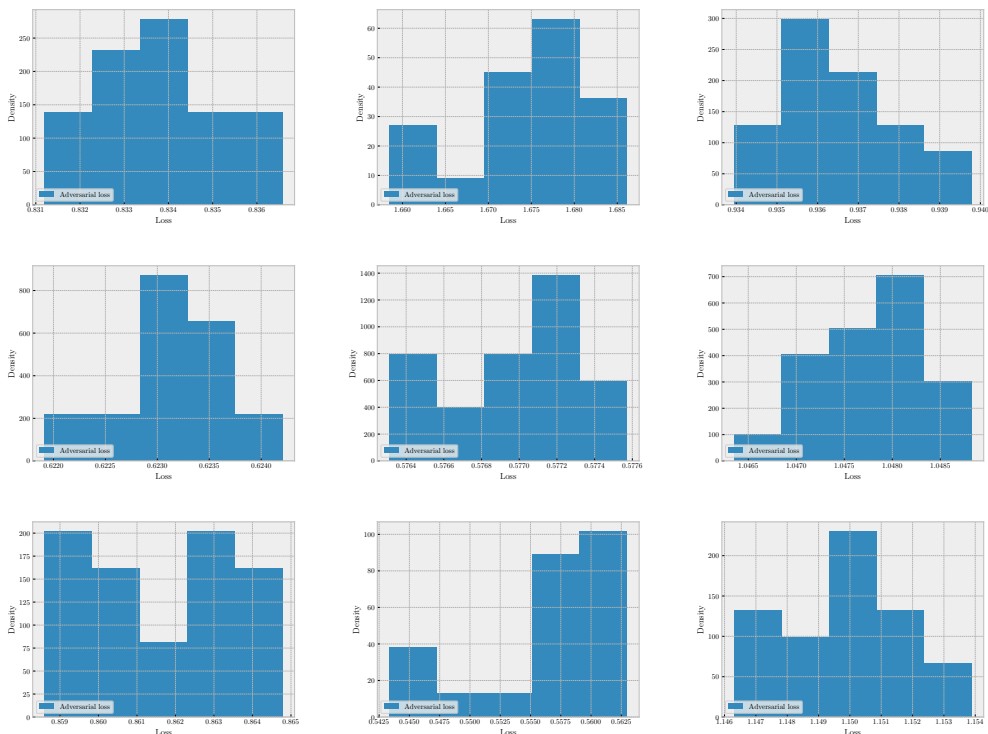

Figure 7: The losses of multiple perturbations on 9 different example all concentrate around a value much larger than the clean loss (see Figure 6 for comparison with the clean loss).

## B.2 JUSTIFYING OUR MODIFICATIONS

For Danskin's Theorem Theorem 1 to hold, we require the function to be differentiable. To satisfy differentiability, we replace ReLU with CELU (Barron, 2017) , which has been found to have comparable performance and sometimes outperform ReLU (Dubey et al., 2022).

To operate on individual images and remove the batch-wise correlations across samples we replace BatchNorm (BN) (Ioffe & Szegedy, 2015) with GroupNorm (GN) (Wu & He, 2018)[2].

Finally, to make each update depend only on the current state, we set $momentum = 0.0$. Since momentum is standard practice in the CV community and works like Yan et al. (2018) argue that it can improve generalisation, we rely on our ablation to show that removing it is safe.

## B.3 FURTHER DETAILS ON SYNTHETIC EXPERIMENTS

The synthetic experiment in Fig. 1a is conducted with the following settings. The inner-maximization is approximated with 10 steps of projected gradient ascent in order to match the traditional AT setting. The outer iterations have a decaying $\frac{0.5}{\sqrt{k}}$ step-size schedule. We observe the same erratic behavior for PGD with a fixed outer stepsize, while DDi consitently remains well-behaved.

The synthetic experiment in Fig. 3 is conducted on a dataset of size 100 in dimension 2 where the coordinates are standard Gaussian. The neural network is a 2-layer network with `ELU` activation with a hidden layer of width 2. The inner solver is PGD with 10 steps with stepsize 0.1 and optimizes

---

[2] There are whole lines of work studying the effects of BN (Bjorck et al., 2018; Santurkar et al., 2018; Kohler et al., 2019) as well as removing it altogether(Brock et al., 2021). It has also been found to interact with adversarial robustness in Wang et al. (2022) and Benz et al. (2021), the latter also finds GN to be a well performing alternative, justifying our choice.

over the unit cube. The outer step-size is $0.01$ and the weights are optimized with full batch gradient descent.

The linear approximation at iteration $250$ of the robust loss consits of taking the $10$ adversarial examples computed at iteration $250$ and approximating it with

$$\tilde{\phi}(\theta) = \max_{\delta_1 \dots \delta_{10}} \phi(\theta_{250}) + \langle \nabla_\theta g(\theta_{250}, \delta_i), \theta - \theta_{250} \rangle$$

Interestingly we do not observe the same drastic improvement over PGD when observing the non-linearized loss at iteration $250$.

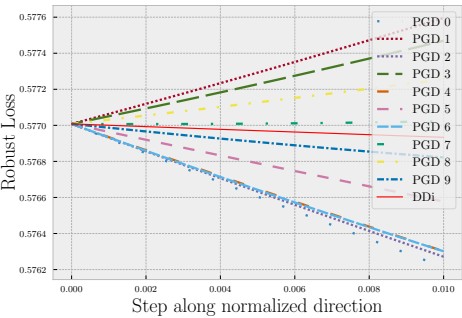

## B.4  IMPROVING THE RUNNING TIME

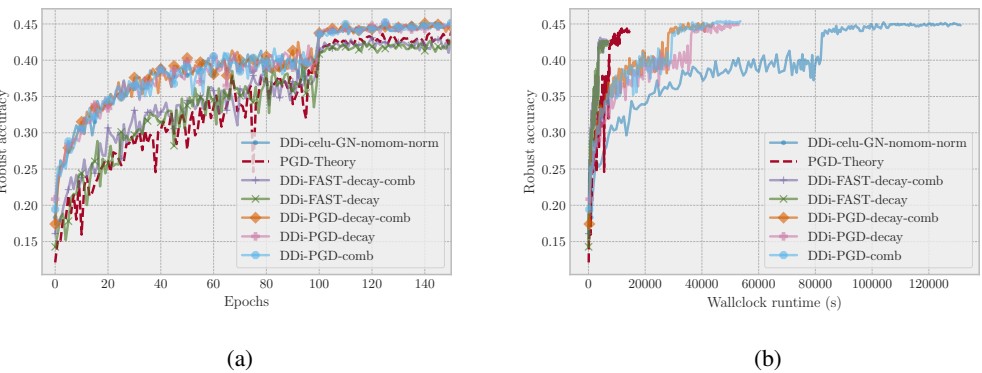

|  (a)  |  (b)  |

Figure 8: (a) Epoch evolution of a more efficient implementation of DDi. (b) Wallclock evolution of the same methods.

While the focus of this paper is not to obtain a state-of-the-art method, it does matter whether it is feasible to efficiently capture the benefit of DDi. The naive implementation has about a $10 - 12$ times overhead compared to PGD, mainly due to three bottlenecks (in descending impact)

1. for $k$-DDi, generating $k$ adversarial examples with PGD as the base attack involves a $k$-times overhead

2. then $k$ separate gradient samples need to be computed on these adversarial examples, which involes $k$ forward-backward passes

3. finally, one additional optimization problem needs to be solved.

While steps 1) and 2) can be somewhat parallelized, they still cause a massive increase in compute and memory. We therefore adopt two heuristic approaches to speed up the algorithm while (hopefully) maintaining it's benefits:

1. since later in training the benefits of DDi appear to diminish, we linearly decay the number of gradients sampled $k$ from 10 down to 1 along the 200 epochs (referred to as `decay`)

    2. we also adopt a method of creating $k$ unique batches from only 2 independent adversarial attacks (described below in Appendix B.4.1, referred to as `comb`).

We evaluate this method using both PGD and FAST as base attacks and show the results in Fig. 8a and Fig. 8b. As can be seen, `DA-PGD-decay-comb` and `DA-PGD-comb` both enjoy a massive speedup in wallclock time (reducing the $12\times$ overhead to about $3\times$) while retaining the improved per-step progress of base $DDi$.

### B.4.1 COMBINATORIAL BATCH CONSTRUCTION

Suppose we have a batch of data-label pairs $(x_i, y_i)$ of size $B$. In order to construct $k \leq 2^B$ different gradients by computing only 2 adversarial examples per data sample $x_i$ in the batch we do the following:

    1. for each $i = 1, \ldots, B$ compute $\delta_{i,0}, \delta_{i,1}$ two adversarial examples using the data-label pair $(x_i, y_i)$ in the batch.

    2. for each $j = 1, \ldots, k$ repeat the following steps:

    3. Define $\Delta = [\,]$ as an empty list.

    4. generate a random bitvector $b \subseteq \{0, 1\}^B$ of length $B$

    5. when $b_i$ is 0 we append $\delta_{i,0}$ to $\Delta$, otherwise when $b_i$ is 1 we append $\delta_{i,1}$ to $\Delta$.

    6. compute the gradient w.r.t. the network parameters using the perturbations in $\Delta$

While this still incurs overhead of computing $k$ gradients, it greatly reduces running time as seen in Fig. 8b and could further improved by e.g. reusing gradients from past epochs to construct the examples.

## C PROOF OF THEOREM 2.

The steepest descent direction is computed, following Eq. (4) as:

$$\gamma^\star \in \underset{\gamma:\|\gamma\|_2=1}{\arg\min} D_\gamma\phi(\theta) = \underset{\gamma:\|\gamma\|_2=1}{\arg\min} \max_{\delta \in \mathcal{S}_m^\star(\theta)} \langle \gamma, \nabla_\theta g(\theta, \delta) \rangle \tag{15}$$

Whenever $\theta$ is not a local optimum, there exists a non-zero descent direction. In this case we can relax the constraint that $\|\gamma\|_2 = 1$ to $\|\gamma\|_2 \leq 1$ without changing the solutions or optimal value of (15), which is strictly negative:

$$\min_{\gamma:\|\gamma\|_2=1} \max_{\delta\in\mathcal{S}_m^\star(\theta)} \langle \gamma, \nabla_\theta g(\theta, \delta) \rangle = \min_{\gamma:\|\gamma\|_2\leq1} \max_{\delta\in\mathcal{S}_m^\star(\theta)} \langle \gamma, \nabla_\theta g(\theta, \delta) \rangle < 0 \tag{16}$$

We can now transform (15) into a bilinear convex-concave min-max problem, subject to convex and compact constraints:

$$\begin{aligned}
\gamma^\star \in \underset{\gamma:\|\gamma\|_2\leq1}{\arg\min} D_\gamma\phi(\theta) &= \underset{\gamma:\|\gamma\|_2\leq1}{\arg\min} \max_{\delta\in\mathcal{S}_m^\star(\theta)} \langle \gamma, \nabla_\theta g(\theta, \delta) \rangle \\
&= \underset{\gamma:\|\gamma\|_2\leq1}{\arg\min} \max_{i=1,\ldots,m} \gamma^\top \nabla_\theta g(\theta, \delta^{(i)}) \\
&= \underset{\gamma:\|\gamma\|_2\leq1}{\arg\min} \max_{\alpha\in\Delta^m} \gamma^\top \nabla_\theta g(\theta, \mathcal{S}_m^\star(\theta))\alpha
\end{aligned} \tag{17}$$

By Sion's minimax Theorem Sion (1958), we can solve Eq. (17) by swapping the operator order:

$$\begin{aligned}
\min_{\gamma:\|\gamma\|_2\leq1} \max_{\alpha\in\Delta^m} \gamma^\top \nabla_\theta g(\theta, \mathcal{S}_m^\star(\theta))\alpha &= \max_{\alpha\in\Delta^m} \min_{\gamma:\|\gamma\|_2\leq1} \gamma^\top \nabla_\theta g(\theta, \mathcal{S}_m^\star(\theta))\alpha \\
&= \max_{\alpha\in\Delta^m} -\|\nabla_\theta g(\theta, \mathcal{S}_m^\star(\theta))\alpha\|_2 \\
&= -\min_{\alpha\in\Delta^m} \|\nabla_\theta g(\theta, \mathcal{S}_m^\star(\theta))\alpha\|_2 < 0
\end{aligned} \tag{18}$$

Finally, by noting that squaring the objective function in the right-hand side of Eq. (18) does not change the set of solutions, we arrive at the formula for $\alpha^\star$ in Eq. (14). Indeed for a solution $\alpha^\star$ to

this problem we have

$$
\begin{aligned}
\arg\min_{\gamma:\|\gamma\|_2 \leq 1} \max_{\alpha \in \Delta^m} \gamma^\top \nabla_\theta g(\theta, \mathcal{S}_m^\star(\theta))\alpha &= \arg\min_{\gamma:\|\gamma\|_2 \leq 1} \gamma^\top \nabla_\theta g(\theta, \mathcal{S}_m^\star(\theta))\alpha^\star \\
&= -\frac{\nabla_\theta g(\theta, \mathcal{S}_m^\star(\theta))\alpha^\star}{\|\nabla_\theta g(\theta, \mathcal{S}_m^\star(\theta))\alpha^\star\|}
\end{aligned}
\tag{19}
$$

where the denominator is nonnegative as the optimal objective value is nonzero c.f. Eq. (18).

## D  PROOF OF THEOREM 3.

For any $\delta \in \mathcal{S}^\star(\theta)$ let $i(\delta) \in \{1, \ldots, m\}$ be such that $\|\delta^{(i(\delta))} - \delta\|_2 \leq \epsilon$. That is, we map any maximizer $\delta$ to an index $i \in \{1, \ldots, m\}$ such that the corresponding perturbation $\delta^{(i)}$ in the finite set $\mathcal{S}_m^\star(\theta)$ is at most at an $\epsilon$ distance. This map can be constructed by the assumption on $\mathcal{S}_m^\star(\theta)$.

For any $\gamma$ such that $\|\gamma\|_2 = 1$ we have

$$
\begin{aligned}
\langle \gamma, \nabla_\theta g(\theta, \delta) \rangle &= \langle \gamma, \nabla_\theta g(\theta, \delta) - \nabla_\theta g(\theta, \delta^{(i(\delta))}) \rangle + \langle \gamma, \nabla_\theta g(\theta, \delta^{(i(\delta))}) \rangle \\
&\leq \underbrace{\|\gamma\|_2}_{=1} \underbrace{\|\nabla_\theta g(\theta, \delta) - \nabla_\theta g(\theta, \delta^{(i(\delta))})\|}_{\leq L\|\delta - \delta^{(i(\delta))}\| \leq L\epsilon} + \langle \gamma, \nabla_\theta g(\theta, \delta^{(i(\delta))}) \rangle \\
&\leq \langle \gamma, \nabla_\theta g(\theta, \delta^{(i(\delta))}) \rangle + L\epsilon \\
&\leq \sup_{\delta \in \mathcal{S}_m^\star(\theta)} \langle \gamma, \nabla_\theta g(\theta, \delta^{(i)}) \rangle + L\epsilon
\end{aligned}
\tag{20}
$$

Taking the supremum over $\delta \in \mathcal{S}^\star(\theta)$ on the left-hand-side we obtain

$$
D_\gamma \phi(\theta) \coloneqq \sup_{\delta \in \mathcal{S}^\star(\theta)} \langle \gamma, \nabla_\theta g(\theta, \delta) \rangle \leq \sup_{\delta \in \mathcal{S}_m^\star(\theta)} \langle \gamma, \nabla_\theta g(\theta, \delta^{(i)}) \rangle + L\epsilon
\tag{21}
$$

Hence if the supremum on the right-hand-side is strictly smaller than $-L\epsilon$ we have that $D_\gamma \phi(\theta) < 0$, which yields the desired result.

## E  PROOF OF LEMMA 1

Assume the limit that defines $\hat{D}_\gamma \phi(\theta)$ exists (and is finite).

$$
\begin{aligned}
\hat{D}_{-\gamma} \phi(\theta) &= \lim_{t \to 0} \frac{\phi(\theta + t(-\gamma)) - \phi(\theta)}{t\|-\gamma\|_2} \\
&= \lim_{t \to 0} \frac{\phi(\theta + (-t)\gamma) - \phi(\theta)}{-(-t)\|\gamma\|_2} \\
&= \lim_{(-t) \to 0} \frac{\phi(\theta + (-t)\gamma) - \phi(\theta)}{-(-t)\|\gamma\|_2} \\
&= \lim_{s \to 0} -\frac{\phi(\theta + s\gamma) - \phi(\theta)}{s\|\gamma\|_2} \qquad \text{(let } s = (-t)) \\
&= -\lim_{s \to 0} \frac{\phi(\theta + s\gamma) - \phi(\theta)}{s\|\gamma\|_2} \\
&= -\hat{D}_\gamma \phi(\theta)
\end{aligned}
\tag{22}
$$

