# OpenReview forum: "Finding Actual Descent Directions for Adversarial Training"
_ICLR.cc/2023/Conference — ICLR 2023 poster_

### Official Review · Reviewer_wzJU · 2022-10-18

**Confidence:** 4
**Correctness:** 3
**Technical Novelty And Significance:** 4
**Empirical Novelty And Significance:** 3
**Recommendation:** 8

**Clarity, Quality, Novelty And Reproducibility:**

The paper is well-written, the content is well-presented and organized. To the best of my knowledge, the problem the authors point out is not well-known in the adversarial training literature, for which the presented solution is novel.

**Strength And Weaknesses:**

The authors raise awareness about a mistaken interpretation of Danskin's Theorem when motivating adversarial training, and demonstrate its relevance through counter-examples and empirical evidence. A solution to this problem is presented (DDD), and its relative effectiveness on smooth networks (without BatchNorm) is shown.
At this stage, the contribution is purely technical: the practical relevance of the presented method is limited because of its inferior performance in the setting that attains the maximal robust accuracy (ReLU + BatchNorm). Furthermore, DDD greatly increases the runtime per iteration, making its practical employability limited.
Nevertheless, I believe the findings to be of great interest to the community.

In my opinion, the strength of the paper could benefit if the authors addressed the following limitations:
- Typically, in the context of adversarial training, the inner maximization is not run to convergence. Indeed, this holds also for the presented experiments. Therefore, DDD is not guaranteed to yield a descent direction for the robust loss even when the conditions of Danksin's Theorem hold. It would be important for the authors to comment on why DDD helps in this setting. Indeed, getting to a guaranteed global optimum of the inner maximization would require running a complete verification algorithm based on Branch and Bound (it would be nice if the authors touched on this, and mentioned relevant methods in the literature).
- Practical limitations (in practice the overall best-performing method remains PGD) should be stated in intro and abstract (as the authors do in the conclusions) . The statement on the early stages should be limited to the smooth setup (the difference in Figure 4b seems to be too small to be statistically reliable).

As a minor point: why not Frank-Wolfe for the smooth simplex-constrained optimization problem that arises from DDD?
Finally, the fact that the inner maximization in AT is not run to global optimality would be visibile if the authors provided a magnified version of the high-loss region of Figure 2.

**Summary Of The Paper:**

The authors point out that the interpretation of Danskin's Theorem that was used to motivate the Adversarial Training paradigm in the seminal paper from Madry et al. (2018) relies on a misunderstanding of the notion of directional derivative employed in the theorem. As a consequence, PGD, the most commonly employed adversarial training scheme, might not follow descent directions. The authors present a solution to this, assuming finitely-many global optimizers of the inner maximization, and show that it helps in practice in setttings where Danksin's Theorem would hold.

**Summary Of The Review:**

The authors point out an interesting problem in the adversarial training literature: while the practical effectiveness of the presented method is limited at best, and the presented solution is never used in the setting for which it is motivated (the inner maximization is never run to convergence, as that would require running formal neural network verification), I believe the paper is a first step towards more principled adversarial training algorithms.
Nevertheless, the authors should be more direct about the limitations of the proposed solution and its empirical relevance, from the beginning of the paper. I will be willing to increase my score after this and the other weaknesses I pointed out are addressed.

---

> ### Author Response · Authors · 2022-11-14
> **Addresing your concerns + summary of changes**
>
> We have uploaded a new version of the paper with changes based on your feedback, for which we are thankful. We believe that this has made our paper stronger. We are hopeful that the new changes resolve your concerns, thus improving your opinion of our work and maybe making you consider raising the score. All essential changes in the new version are highlighted in blue color. We summarize the changes that were inspired by your observations, and answer your concerns:
>
> > **Q1.** In the context of adversarial training, the inner maximization is not run to convergence. Therefore, DDD is not guaranteed to yield a descent direction for the robust loss even when the conditions of Danksin's Theorem hold. It would be important for the authors to comment on why DDD helps in this setting.
>
> **A1.** In order to address this, we have added a new theoretical result (Theorem 3 in page 5) that provides a way to certify a descent direction even if the set of maximizers $S^\star(\theta)$ is infinite and not available. We instead assume a finite set $S^\star_m(\theta)=\{\delta^{(1)}, \ldots, \delta^{(m)}\}$ that **doesn't need to be composed of maximizers** (hence the inner-maximization does not need to be solved to optimality). Instead we just assume that $S^\star_m(\theta)$ is a good approximation of $S^\star(\theta)$ in the sense that for all $\delta \in S^\star(\theta)$ there should be a $\delta^{(i)} \in S^\star_m(\theta)$ such that $\| \delta - \delta^{(i)} \| \leq \epsilon$. We believe this is a more reasonable assumption as one can expect that by repeatedly running PGD with a large number of iterations, one should be able to "cover" the set of solutions up to some small distance $\epsilon$. Of course, certifying that a finite set satisfies this property for some $\epsilon$ is a hard problem, but we stress the fact that, given the non-concavity and generality of the maximization problem, at some point we need to make a strong assumption to derive results. Hopefully you will agree that this assumption is more reasonable.
>
> > Q2. Indeed, getting to a guaranteed global optimum of the inner maximization would require running a complete verification algorithm based on Branch and Bound (it would be nice if the authors touched on this, and mentioned relevant methods in the literature).
>
> **A2.** In the paragraph following the new Theorem 3 we have mentioned this issue, and how the less restrictive result of Theorem 3 does not require optimality of the inner maximization. We have added the following references [A] Evaluating Robustness of Neural Networks with Mixed Integer Programming. Tjeng et al. ICLR 2019 and [B] A Branch and Bound Framework for Stronger Adversarial Attacks of ReLU Networks. Zhang et al. ICML 2022. If we are missing some reference please let us know.
>
> > Q3. In practice the overall best-performing method remains PGD, should be stated in intro and abstract, as the authors do in the conclusions. The statement on the early stages should be limited to the smooth setup (the difference in Figure 4b seems to be too small to be statistically reliable).
>
> **A3.** We have modified the abstract and introduction as you suggest. We remark that in the non-smooth setup + BatchNorm there is no theory that explains this behaviour as Danskin's theorem does not even apply, and the function is not even differentiable due to the presence of ReLUs.
>
> > Q4. why not Frank-Wolfe for the smooth simplex-constrained optimization problem that arises from DDD?
>
> **A4.** Frank-Wolfe is indeed a valid approach to solve the inner problem but its worst case convergence rate is $1/k$ whereas our current method, accelerated projected gradient descent can attain the optimal $1/k^2$ rate.
>
> > Q5. The fact that the inner maximization in AT is not run to global optimality would be visibile if the authors provided a magnified version of the high-loss region of Figure 2.
>
> **A5.** As you suggest, In appendix B we have added magnified versions of the high-loss regions of Figure 2.
>
> We remain at your disposal to answer any further questions.

---

> > ### Comment · Reviewer_wzJU · 2022-11-18
> > **Thanks for your response**
> >
> > I thank the authors for their response, which addressed most of my concerns. I increased my score to 8.
> >
> > A couple of small points:
> >
> > **A2.** It would be nice if the authors could cite further state-of-the-art algorithms based on branch-and-bound. These could include:
> > [C] Scaling the Convex Barrier with Active Sets, De Palma et al, ICLR 2021;
> > [D] Beta-CROWN: Efficient Bound Propagation with Per-neuron Split Constraints for Complete and Incomplete Neural Network Robustness Verification, Wang et al, NeurIPS 2021.
> >
> > **A4.** I might be mistaken, but isn't the problem strongly convex? I think Frank-Wolfe variants such as away and pairwise steps should have a linear convergence rate in this setting (and so does PGD). But for FW variants, one should be able to determine the optimal step size analytically.

---

> > > ### Author Response · Authors · 2022-11-21
> > > **Reply to further feedback**
> > >
> > > Dear Reviewer,
> > >
> > > thank you for reading our rebuttal and re-evaluating your score.
> > >
> > > **Regarding A2.** We will add the mentioned missing references our next revision.
> > >
> > > **Regarding A4.** The problem **may be** strongly convex depending on the matrix $\nabla_\theta g(\theta, S^\star_m(\theta))$. We would need all columns to be linearly independent and this not true in general so **strong convexity is not guaranteed**. Moreover, it is known that vanilla FW has slower rates when the solution is not in the interior of the constraint set. In our experience the solutions are usually not in the interior of the simplex but rather at the boundary, hence the **vanilla** FW algorithm would have a slower sublinear $1/t$ convergence rate even if the obj. is str. convex. As you mention the addition of away steps (or other modifications) can alleviate this issue and get improved rates but this requires a more careful study of the properties of our objective function: known FW results require very precise assumptions.
> > >
> > > Moreover, note that the LMO for the simplex has linear complexity whereas the efficient projection on the simplex is only $\mathcal{O}(n \ln(n))$ so it would be a minor speed improvement that removes the logarithmic term. In a new revision we will mention that FW with away steps is an alternative way to obtain $\alpha^\star$. Currently it is not clear which method would be faster in practice. We might check experimentally in a future revision. Thank you anyways for the suggestion of FW+AS!

---

### Official Review · Reviewer_FpN3 · 2022-10-25

**Confidence:** 4
**Correctness:** 4
**Technical Novelty And Significance:** 4
**Empirical Novelty And Significance:** 4
**Recommendation:** 10

**Clarity, Quality, Novelty And Reproducibility:**

**Clarity.**  The paper was easy to read.

**Quality.**  The results were technically sound, and the experiments were thorough.

**Novelty.**  The algorithm seems novel, and this work offers a fresh perspective on adversarial training.

**Reproducibility.**  This seems relatively easy to reproduce based on the details given in the paper.

**Strength And Weaknesses:**

### Strengths

**Soundness.**  This paper seems to be technically quite sound.  It is certainly the most rigorous paper that I have reviewed in my batch of papers this year at ICLR.  The analysis of Danskin's theorem and of the result in (Madry et al., 2018) appears to be correct, which indeed calls into question one of the foundational assumptions underpinning adversarial training.

The counterexamples shed new light on the problem as well.  I particularly appreciated the fact that after disproving the theorem from (Madry et al., 2018), the authors took it one step forward.  They ask: In the case when I'm not at a locally optimal point, is the result of (Madry et al., 2018) still incorrect.  And via a more intricate, although still pleasingly elementary, construction, they show that another counterexample exists.  This is thorough work, and I believe that it should be commended.

**Algorithm.**  The algorithm proposed by the authors seemed relatively natural.  While it is clearly up for debate whether the set of maximizers is finite, I think it's not an unreasonable assumption to make.  And the authors provide some nice empirical evidence that at the very least, the set is not a singleton.  Given this, Thm. 2 makes sense given the context of the paper, and it leads to a relatively practical algorithm.  Nice!

**Well written.**  This paper is relatively well written.  The related work was thorough.  And in general, the logic and flow of the paper was solid.  I really enjoyed reading it!

### Weaknesses

**Minor inconsistencies in notation.**  One point of confusion: if $\delta\in\mathbb{R}^p$, then $\mathcal{S}\subset\mathbb{R}^p$ in (1).  In (2), we have $\delta_i\in\mathbb{R}^p$, and therefore $\delta$ seems to be an element of $\mathbb{R}^{p\times k}$, meaning that $\mathcal{S}\subset \mathbb{R}^{p\times k}$ (or perhaps of $\mathbb{R}^{pk}$).  This is a bit inconsistent, as it would be nice to think of $\mathcal{S}$ as belonging to a particular space.

**Confusing presentation of the key theorem.**  One key step of this paper is to present the theorem of (Madry et al., 2018), which is done in Corollary 1.  Before presenting this result, the authors say that

> "Corollary 1 is an equivalent rephrasing of Madry et al. (2018, Corollary C.2.), and is derived as a consequence of Theorem 1.
 Unfortunately counterexample 1 shows that it is false:"

Let Thm. 1 (Danskin's theorem) be A and let Cor. 1 be B.  Based on my reading, this sentence says that A implies B, since "derived" (to me) means "correctly derived."  The authors then say that B is false.  By the contrapositive, this would imply that A is false.  Based on the rest of the paper, I know that this is not what the authors are trying to say.  So I would suggest rewording this part to emphasize that A does not imply B, as some readers may get confused by this point, and that is quite undesirable given that this is one of the most important points in the paper.

I would also recommend defining a "descent direction" before it is mentioned in Cor. 1.  This would make it easier to understand the result in Cor. 1 without having to read on until a later section.

**A minor recommendation.**  I would recommend removing the phrase: "a basic concept in multivariable calculus."  This may only be my opinion, but I feel that this phrase may be perceived by some readers as a little bit disrespectful to (Madry et al., 2018), especially given that (Madry et al., 2018) is a relatively influential paper.  Yes, this idea is often introduced in a college-level calculus course.  But it took until now for anyone to notice this subtle distinction.

**Explain Madry et al.'s results in more detail.**  I think it's well worth spending more time on the results of the original paper.  The discussion at the bottom of page 3 is relatively terse.  In particular, I believe that it would be beneficial for the authors to expand on the sentence:

> "However, we cannot guarantee that the function will decrease if we move in the opposite direction..."

During my first pass through the paper, I didn't understand this part.  Underscoring the fact that Madry et al. assume that $D_\gamma \phi(\theta) > 0$ *implies that* $D_{-\gamma} \phi(\theta) < 0$ is essential here, because it is at the heart of why the original result was not correct.  It may be worth proving that for the two-sided directional derivative, this property does hold.  Although elementary, I think this would help build intuition for the reader.

**Oracle?**  I didn't understand what the authors meant by a "heuristic oracle."  The word "heuristic" seems to defeat the purpose of an oracle.  Could the authors explain more here?

**Computation time.**  A downside of the proposed method is obviously the computation time.  It seems clear that at convergence, DDD doesn't offer a significant improvement over PGD given that it seems to get 10x more steps.  I may have missed this, but did the authors compare PGD-100 to DDD-10, which would constitute a comparison where each algorithm got roughly the same computational budget?

**Summary Of The Paper:**

The starting point for this paper is the observation that one of the key theorems in (Madry et al., 2018) is incorrect.  In particular, the authors show that the interpretation of Danskin's theorem in (Madry et al., 2018) uses the incorrect form of the directional derivative.  They provide several counterexamples to demonstrate this flaw, which yield interesting insights on the problem of adversarial training.

Based on these observations, the authors propose Danskin's Descent Direction (DDD), an algorithm that is designed to solve the standard minimax formulation of adversarial training.  They show that their method is more stable than PGD on several benchmarks, although the empirical gains are somewhat modest at convergence.

**Summary Of The Review:**

Overall, I thought this was a really solid paper.  It makes some interesting insights, and the result is an algorithm that seems to have some nice empirical properties.  There are a few drawbacks, such as the increased computation time.  Most of the weaknesses listed above are relatively minor.  Therefore, I think this paper should be accepted.

---

**Post rebuttal comments.**  The authors have addressed each of my concerns.  Therefore, I will raise my score, as I believe that the additional clarifications and theoretical results improve the paper.

---

> ### Author Response · Authors · 2022-11-14
> **Addressing your concerns + summary of changes (Part 2 of 2)**
>
> > **Q5.** I would recommend removing the phrase: "a basic concept in multivariable calculus." This may only be my opinion, but I feel that this phrase may be perceived by some readers as a little bit disrespectful to (Madry et al., 2018)
>
> **A5.** We have removed the word "basic". We stress that it was never our intention to appear disrespectful to other authors.
>
> > **Q6.** During my first pass through the paper, I didn't understand this part. Underscoring the fact that Madry et al. assume that $D_\gamma \phi(\theta) > 0$ implies that
> $D_{-\gamma}(\theta) < 0$ is essential here, because it is at the heart of why the original result was not correct. It may be worth proving that for the two-sided directional derivative, this property does hold. Although elementary, I think this would help build intuition for the reader.
>
> **A6.** We have re-stated this property as lemma 1 (end of page 3) and we provide a proof in appendix E for the reader interested in the details.
>
> > **Q7.** I didn't understand what the authors meant by a "heuristic oracle." The word "heuristic" seems to defeat the purpose of an oracle. Could the authors explain more here?
>
> **A7.** We have rephrased this part. What we mean is that there is a theoretical "oracle" algorithm that finds optimal or near-optimal adversarial perturbations, and such oracle is replaced "in practice" by like repeated runs of PGD. The modifications can be found at the end of page 5.
>
> > **Q8.** A downside of the proposed method is obviously the computation time. It seems clear that at convergence, DDD doesn't offer a significant improvement over PGD given that it seems to get 10x more steps. I may have missed this, but did the authors compare PGD-100 to DDD-10, which would constitute a comparison where each algorithm got roughly the same computational budget?
>
> **A8.** We draw attention to Figure 5.a where we run various attempts of the ablation you request, specifically PGD-70 and PGD-70-1/t. By default, PGD runs with 7 steps, a 10x computational effort corresponds to 70 steps, not 100. To ensure convergence, we run both fixed step size (PGD-70) and a 1/t step size decay once leaving the "default" 7 steps (PGD-70-1/t. As can be seen in the figure, the additional compute does not on its own improve the progress per-step (PGD-70). On the other hand, solving the inner-problem closer to optimality does help (1/t).
>
> We also ran greedy approximations to our algorithm that skip the optimisation algorithm, we either take the gradient corresponding to the *maximum adversarial loss* or the *minimum gradient norm*. We can see that the latter (min-10) compares similarly to PGD-70-1/t, but does not achieve the full benefit of DDD-10. We assume that running a DDD-10-70-1/t variant might further improve performance but requires an even larger compute budget.
>
> Evenmore, we have added a new appendix (appendix B.4) where we explore some modifications to improve the runtime of our algorithm. **As can be seen in Figure 8, such modifications can reduce the runtime of the original DDD by at least 70%. They are still slower than PGD but retain the benefits of DDD in the differentiable activations setting**. We think this provides good evidence that the method could be made practical in a future work.

---

> > ### Comment · Reviewer_FpN3 · 2022-11-14
> > **Response re: summary of changes**
> >
> > Thanks for your clarifications.  They certainly address all of the points raised in my review, and I feel that these changes have made the paper stronger.  In particular, the new result -- Thm. 3 -- strengthens the claims of the paper by looking more closely into whether the assumption of a finite set of minimizers is reasonable.
> >
> > As the authors have improved the paper, I think it's only fair that I improve my score.

---

> ### Author Response · Authors · 2022-11-14
> **Addressing your concerns + summary of changes (Part 1 of 2)**
>
> We have uploaded a new version of the paper with changes based on your feedback, for which we are thankful. We believe that this has made our paper stronger. We are hopeful that the new changes resolve your concerns, thus improving your opinion of our work and maybe making you consider raising the score. All essential changes in the new version are highlighted in blue color. We summarize the changes that were inspired by your observations, and answer your concerns:
>
> > **Q1.** The algorithm proposed by the authors seemed relatively natural. While it is clearly up for debate whether the set of maximizers is finite, I think it's not an unreasonable assumption to make.
>
> **A1.** Other reviewers have raised concerns about the assumption, which we initially justified as a less restrictive assumption than the unique maximizer one (which indeed can be shown to be false via experiments, as we did). For this reason we have added a new theoretical result (Theorem 3 in page 5) that provides a way to certify a descent direction even if the set of maximizers $S^\star(\theta)$ is infinite and not available. We instead assume a finite set $S^\star_m(\theta)=\{\delta^{(1)}, \ldots, \delta^{(m)}\}$ that **doesn't need to be composed of maximizers** (hence the inner-maximization does not need to be solved to optimality). Instead we just assume that $S^\star_m(\theta)$ is a good approximation of $S^\star(\theta)$ in the sense that for all $\delta \in S^\star(\theta)$ there should be a $\delta^{(i)} \in S^\star_m(\theta)$ such that $\| \delta - \delta^{(i)} \| \leq \epsilon$. We believe this is a more reasonable assumption as one can expect that by repeatedly running PGD with a large number of iterations, one should be able to "cover" the set of solutions up to some small distance $\epsilon$. Of course, certifying that a finite set satisfies this property for some $\epsilon$ is a hard problem, but we stress the fact that, given the non-concavity and generality of the maximization problem, at some point we need to make a strong assumption to derive results. Hopefully you will agree that this assumption is even more reasonable.
>
> > **Q2.** One point of confusion: if $\delta \in \mathbb{R}^p$, then $S \subseteq \mathbb{R}^p$ in (1). In (2), we have $\delta_i \in \mathbb{R}^p$ and therefore $\delta$ seems to be an element of $\mathbb{R}^{p \times k}$ meaning that $S \subseteq \mathbb{R}^{p \times k}$. This is a bit inconsistent, as it would be nice to think of $S$ as belonging to a particular space.
>
> **A2.** Indeed this might be confusing as usually $S$ is understood as the constraint for the perturbation of a **single sample** i.e. $S \subseteq \mathbb{R}^p$. However, from an optimization perspective, if we consider the problem over a batch of samples one can simply collect all perturbations for each sample in a new variable $\delta=[\delta_1, \ldots, \delta_k] \in \mathbb{R}^{p\times k}$ and hence we can equivalently understand $S$ as being a subset of $\mathbb{R}^{p \times k}$. We have introduced a sentence clarifying this, immediately after equation (2) in page 3. We now refer to the single perturbation space as $\mathcal{S}_0$.
>
> > **Q3.** Let Thm. 1 (Danskin's theorem) be A and let Cor. 1 be B. Based on my reading, this sentence says that A implies B, since "derived" (to me) means "correctly derived." The authors then say that B is false. By the contrapositive, this would imply that A is false. Based on the rest of the paper, I know that this is not what the authors are trying to say. So I would suggest rewording this part to emphasize that A does not imply B, as some readers may get confused by this point, and that is quite undesirable given that this is one of the most important points in the paper.
>
> **A3.** Following your suggestion we have rephrased this part, and we agree it is expresses more clearly what we mean. The edited text is in page 3 and reads as follows:
>
> > Corollary 1 is an equivalent rephrasing of Madry et al. (2018, Corollary C.2.), and was originally claimed to be a consequence of Theorem 1. Unfortunately counterexample 1 shows that the corollary is false. As Theorem 1 (Danskin’s Theorem) is true, this means that there is some mistake in the proof of the corollary provided in Madry et al. (2018).
>
> This should make our claim more clear.
>
> > **Q4.** I would also recommend defining a "descent direction" before it is mentioned in Cor. 1. This would make it easier to understand the result in Cor. 1 without having to read on until a later section.
>
> **A4.** We have moved the remark defining the concept of descent direction, and is now placed before Cor. 1. We agree this improves the readability.

---

### Official Review · Reviewer_UVpf · 2022-10-25

**Confidence:** 2
**Clarity, Quality, Novelty And Reproducibility:** The paper is written clearly and the …
**Correctness:** 3
**Technical Novelty And Significance:** 3
**Empirical Novelty And Significance:** 3
**Recommendation:** 6

**Strength And Weaknesses:**

Strengths:

The claims of the paper seem solid, and the paper is well written.

The proposed method is backed up by experiments.

Weaknesses:

Theorem 2 is based on the assumption that the set of optimal adversarial perturbations is finite. This is a strong and sometimes unrealistic assumption.

According to the authors' discussion, it seems that Madry et al. (2018, Corollary C.2) could be made rigorous by adding an assumption such as "$\phi(\theta)$ is differentiable at $\theta$". Even with this additional assumption, this result seems sufficient to motivate the adversarial training algorithm.

Because of the above reasons, the advantage of the proposed DDD method over the adversarial training method is not very clear. After all, the theoretical guarantee for the proposed DDD method (Theorem 2) is also based on an unrealistic assumption. The DDD method also seems to be computationally more expensive than adversarial training.

**Summary Of The Paper:**

This paper points out an issue in a recent theoretical result that motivates the adversarial training algorithm. The authors construct counter examples and provide explanations about the issue they identify, and then further propose a Danskin's Descent Direction for training robust neural networks.

**Summary Of The Review:**

This paper points out an important issue in adversarial training. The paper is well written and the claims seem solid. However, the proposed method is still questionable and more justification is needed to demonstrate its advantage over existing methods.

---

> ### Author Response · Authors · 2022-11-14
> **Addressing your concerns + summary of changes**
>
> We have uploaded a new version of the paper with changes based on your feedback, for which we are thankful. We believe that this has made our paper stronger. We are hopeful that the new changes resolve your concerns, thus improving your opinion of our work and maybe making you consider raising the score. All essential changes in the new version are highlighted in blue color. We summarize the changes that were inspired by your observations, and answer your concerns:
>
> > **Q1.** Theorem 2 is based on the assumption that the set of optimal adversarial perturbations is finite. This is a strong and sometimes unrealistic assumption.
>
> **A1.** Our assumption is currently the less strong (less unrealistic) in the literature that leads to descent directions. As we show, the corollary in the original paper by Madry et. al is False. Hence, prior to our work, the only way to guarantee a descent direction for AT was to assume a **unique minimizer** which is much stronger, definitely unrealistic and observably wrong, as we show in the experiments.
>
> Nevertheless, we relax this condition even further: we have added a new theoretical result (Theorem 3 in page 5) that certifies a descent direction even if the set of maximizers $S^\star(\theta)$ is infinite. We instead assume a finite set $S^\star_m(\theta)=\{\delta^{(1)}, \ldots, \delta^{(m)}\}$ that **doesn't need to be composed of maximizers** (hence the inner-max does not need to be solved to optimality). Instead we assume that $S^\star_m(\theta)$ is a good approximation of $S^\star(\theta)$ in the sense that for all $\delta \in S^\star(\theta)$ there exists $\delta^{(i)} \in S^\star_m(\theta)$ such that $\| \delta - \delta^{(i)} \| \leq \epsilon$. This is a more reasonable as repeatedly running PGD with a large number of iterations, one should be able to "cover" the set of solutions up to some small distance $\epsilon$. Of course, certifying that a finite set satisfies this property for some $\epsilon$ is a hard problem, but given the non-concavity and generality of the maximization problem, at some point we need to make a strong assumption to derive results.
>
> > **Q2.** According to the authors discussion, it seems that Madry et al. (2018, Corollary C.2) could be made rigorous by adding an assumption such as $\phi(\theta)$ is differentiable at $\theta$. Even with this additional assumption, this result seems sufficient to motivate the adversarial training algorithm.
>
> **A2.** Adding strong assumptions like differentiability or uniqueness of a maximizer can make Corollary C.2. become true. However (1) this defeats the purpose of such a result, as it becomes trivial and (2) such strong assumptions do not hold even for the partial maximization function $\phi$ corresponding to simple neural networks. In constrast, our assumptions are much less restrictive. We also remark that the case were the maximizer is unique (and hence $\phi$ is differentiable) is usually stated as part of the original 1960's theorem by Danskin.
>
> > **Q3.** The DDD method also seems to be computationally more expensive than adversarial training.
>
> **A3.** Indeed, as our results and the theory suggests, it is computationally hard to get a descent direction for a **highly non-concave** inner maximization problem with possibly infinite maximizers. This is not surprising as such problems are NP-hard in general. Our DDD method is developed to show that the traditional Adversarial Training method might not obtain descent directions at each iteration. However, we believe the research community will benefit from our improved understanding of the method, and that we or someone else, in time, will develop improved methods to drive down the computational complexity of our proposed method, making it more applicable. To provide some evidence, we have added a new appendix (appendix B.4) where we explore some modifications to improve the runtime of our algorithm. **As can be seen in Figure 8, such modifications can reduce the runtime of the original DDD by at least 70%. They are still slower than PGD but retain the benefits of DDD in the differentiable activations setting**. We think this provides good evidence that the method could be made practical in a future work.
>
> > **Q4.** the proposed method is still questionable and more justification is needed to demonstrate its advantage over existing methods.
>
> **A4.**  Our main goal is not to obtain a state-of-the-art method, but to bring new knowledge to the community. We derive our method from principled theoretical results appear to be unknown in the AT community. We believe that heuristic improvements over the vanilla DDD method that we propose here are possible, but are left for future work. Again, **we have added appendix B.4 where we explore some modifications to make the method much faster while retaining the theoretical benefits, showing that indeed the method could be made even faster in future work.**
>
> We remain at your disposal to answer any further questions.

---

### Official Review · Reviewer_gFkW · 2022-10-28

**Confidence:** 4
**Correctness:** 4
**Technical Novelty And Significance:** 3
**Empirical Novelty And Significance:** 3
**Recommendation:** 6

**Clarity, Quality, Novelty And Reproducibility:**

Quality:
The paper is very well written. Delivers a simple yet interesting observation clearly. The observation is to the best of my knowledge novel in the literature of AT.

Clarity:
The paper is well organized. The constructive counter-examples are well explained and the authors have done a good job maintaining their simplicity.

Originality: The main claim and the proposed methods are novel. I appreciate the paper being bold and challenging the mainstream practices.

**Strength And Weaknesses:**

Strength:

I appreciate the conceptual idea of this paper a lot. The starting point of the paper is the observation that the descent direction not necessarily corresponds to the worst-case perturbation. Given the fact that this is typically assumed always be the case in the AT community, this paper has provided a refreshing perspective on how should we do AT correctly. The constructed counterexamples are also easy to understand.

Weakness:

The potential downside of this paper is that the proposed method (DDD) is inherently computationally expensive. As the authors advocate the usage of multiple worst-case perturbations, adopting DDD would require at least several times of computational effort compared to the already expensive standard AT procedure.

Another concern that I have is the similar final performance achieved by both AT and DDD. It seems that their performance differs by some margin only in the initial stage of the training, but is getting much closer to the final stage. This to some extent, has limited the potential practical impact of the proposed method.




**Summary Of The Paper:**

This paper pays attention to the computation of adversarial training, by pointing out that even in the simple case (nonsmooth), the descent direction is not given by the worst-case perturbation, as opposed to common practice in the AT community. The paper then proposes a method that, under the assumption of a finite number of worst-case perturbations, computes the correct descent direction.

**Summary Of The Review:**

This paper focuses on the bring forward the issue that (a single) adversarial example might not give the descent direction, challenging a common belief adopted in the practice of this field. It did this with simple examples making its arguments. This is the strongest point of this paper. The proposed method, on the other hand, is computationally expensive compared to the original AT, and the final performance is comparable to AT, which is to some extent surprising given the fact that AT is supposedly not a correct method, the main theoretical claim made by this paper. The practical side of the claim and method proposed in this paper seems a little bit limited by its current shape.

---

> ### Author Response · Authors · 2022-11-14
> **Addressing your concerns + summary of changes**
>
> We have uploaded a new version of the paper with changes based on your feedback, for which we are thankful. We believe that this has made our paper stronger. We are hopeful that the new changes resolve your concerns, thus improving your opinion of our work and maybe making you consider raising the score. All essential changes in the new version are highlighted in blue color. We summarize the changes that were inspired by your observations, and answer your concerns:
>
> > **Q1.** The potential downside of this paper is that the proposed method (DDD) is inherently computationally expensive. As the authors advocate the usage of multiple worst-case perturbations, adopting DDD would require at least several times of computational effort compared to the already expensive standard AT procedure.
>
> **A1.** We agree that the computational complexity is an issue. However please note that: first, our main contribution is  theoretical, and second, It should be possible to come up with heuristics and modifications to DDD that improve its computational complexity, while retaining some of its benefits (we are in fact working towards such goal). However, such extensions fall outside of the scope of this work. We believe that, in the same way PGD was further made faster through heuristics in multiple papers (Fast-AT, Free-AT, YOPO, etc.) that followed after the original work, the community will greatly benefit from our improved understanding of AT, and possibly improve the method further based on the ideas behind DDD.
>
> To provide some evidence, we have added a new appendix (appendix B.4) where we explore some modifications to improve the runtime of our algorithm. **As can be seen in Figure 8, such modifications can reduce the runtime of the original DDD by at least 70%. They are still slower than PGD but retain the benefits of DDD in the differentiable activations setting**. We think this provides good evidence that the method could be made practical in a future work.
>
> > **Q2.** Another concern that I have is the similar final performance achieved by both AT and DDD. It seems that their performance differs by some margin only in the initial stage of the training, but is getting much closer to the final stage. This to some extent, has limited the potential practical impact of the proposed method.
>
> **A2.** Indeed this is a valid reason of concern. However we stress that our method seems to converge faster (in terms of iterations) so it might be possible that our method improves the speed of convergence rather than the final robustness, which is still an improvement (assuming the iterations could be made faster via some careful modifications). In any case, we believe our main contribution is an improved understanding of the optimization theory behind AT, and this brings value to the ICLR community by convincingly demonstrating new, relevant knowledge.
>
>
> > **Q3.** the final performance is comparable to AT, which is to some extent surprising given the fact that AT is supposedly not a correct method, the main theoretical claim made by this paper.
>
> **A3.** Please note that our main theoretical claim **is not** that AT is an incorrect method. We limit ourselves to claim that the updates of AT might increase, rather than decrease the robustness in a given iteration, and that previously this was not the consensus view in the AT community, given the presence of a false theoretical result in a major influential paper. It might be the case that over multiple iterations, AT still leads to a good value of robustness. There are algorithms like Accelerated Gradient Descent, which do not decrease the objective at each iteration, but still converge to a minimizer. Hence, it might be the case that the advantage of DDD is faster convergence rather than higher robustness.
>
> > **Q4.** The practical side of the claim and method proposed in this paper seems a little bit limited by its current shape.
>
> **A4.** Our work would bring value to the community mostly by demonstrating relevant and impacful theoretical knowledge, and we believe that the community will benefit more from this improved understanding: it can lead to fundamentally different algortithms than current approaches. We think that a practical method using the ideas we present in our work will come with time and further experimentation, in our opinion, such improvements are well-suited to be developed as a separate work. Again, we remark that **we have added appendix B.4 where we explore some modifications to make the method much faster while retaining the theoretical benefits, showing that indeed the method could be made even faster in future work.**
>
> We remain at your disposal to answer any further questions.

---

> > ### Comment · Reviewer_gFkW · 2022-12-07
> > **Thank you for your detailed clarifications**
> >
> > I would like to thank the authors for their detailed response, especially on their efforts to make the proposed method (DDD) a more efficient one compared to its current status. We feel the arguments compelling. My original rating was mostly based on the theoretical observation made by the paper, consequently, I intend to maintain and current score, advocating the acceptance of this paper.

---

### Author Response · Authors · 2022-12-07
**Deadline for discussion period approaching**

Dear reviewers,

as the end of the discussion period is approaching, we would like to thank you again for the valuable discussion and feedback. **If you haven't done so yet**, we would appreciate that you let us know if the changes to the paper have addressed your concerns. In any case, we remain available to answer any further questions.

---

### Decision · Program_Chairs · 2023-01-20

**Decision:**

Accept: poster

**Justification For Why Not Higher Score:**

There is no apparent practical impact of this result yet and the results are known in the optimization community.

**Justification For Why Not Lower Score:**

The paper fixes an error in the justification of adversarial training, which is the main technique to achieve empirical adversarial robustness. Thus this paper contributes to the foundations of this field.

**Metareview: Summary, Strengths And Weaknesses:**

The paper points out an error in the justification of adversarial training using Danskin's theorem when the inner maximization problem has no unique maximum. In this case the negative gradient computed at a particular local maximum need not be a descent direction. The authors illustrate this with examples and provide a new algorithm which is computationally much more expensive.

Strength:
- revealing a wrong argument in the original justification of adversarial training
- provide a fix under the assumption that the set of maximizers is finite
- they show (new theorem 3) that this fix still works for approximate maximizers which "cover" the set of potentially infinite maximizers

Weakness:
- as the authors admit, the results are well-known in the optimization community, so in that sense the result just fixes a wrong argument in a well-known paper
- the suggested alternative DDD is significantly more expensive than adversarial training
- the new method cannot outperform PGD based adversarial training

All reviewers support acceptance, in particular after the authors added the improved result of Theorem 3. While at the moment this paper mainly is a fix to a wrong argument in a well-cited paper, it could inspire subsequent work to improve or better understand adversarial training and thus this is a clear accept.

Detailed comments:
- I could not find details on the computation of robust accuracy, in any case I suggest to use strong attacks like AutoAttack to evaluate the curves in Figure 5 so that there is no doubt that the advantage of DDD is only an evaluation artefact



**Note From Pc:**

if the above contains the word "oral" or "spotlight" please see: "oral" presentation means -> notable-top-5% and "spotlight" means -> notable-top-25%. As stated in our emails, we are disassociating presentation type from AC recommendations